# The Mixture of Probability Distribution Functions for Wind and Photovoltaic Power Systems Using a Metaheuristic Method

Amr Khaled Khamees [1], Almoataz Y. Abdelaziz [2], Makram R. Eskaros [1], Mahmoud A. Attia [3,*]
and Ahmed O. Badr [3]

1  Engineering Physics and Mathematics Department, Faculty of Engineering, Ain Shams University,
   Cairo 11517, Egypt
2  Faculty of Engineering and Technology, Future University in Egypt, Cairo 2430, Egypt
3  Electrical Power & Machines Department, Faculty of Engineering, Ain Shams University, Cairo 11517, Egypt
*  Correspondence: mahmoud.abdullah@eng.asu.edu.eg

**Abstract:** The rising use of renewable energy sources, particularly those that are weather-dependent like wind and solar energy, has increased the uncertainty of supply in these power systems. In order to obtain considerably more accurate results in the analysis of power systems, such as in the planning and operation, it is necessary to tackle the stochastic nature of these sources. Operators require adequate techniques and procedures to mitigate the negative consequences of the stochastic behavior of renewable energy generators. Thus, this paper presents a modification of the original probability distribution functions (PDFs) where the original PDFs are insufficient for wind speed and solar irradiance modeling because they have a significant error between the real data frequency distribution and the estimated distribution curve. This modification is using a mixture of probability distributions, which can improve the fitting of data and reduce this error. The main aim of this paper is to model wind speed and solar irradiance behaviors using a two-component and a three-component mixture of PDFs generated from the integration of the original Weibull, Lognormal, Gamma, and Inverse-Gaussian PDFs. Three statistical errors are used to test the efficiency of the proposed original and mixture PDFs, which are the root mean square error (RMSE), the coefficient of correlation ($R^2$), and the Chi-square error ($X^2$). The results show that the mixture of PDFs gives better fitting criteria for wind speed and solar irradiance frequency distributions than the original PDFs. The parameters of the original and the mixture of PDFs are calculated using the innovative metaheuristic Mayfly algorithm (MA). The three-component mixture of PDFs lowered the RMSE by about 73% and was 17% more than the best original and the two-component mixture distributions.

**Keywords:** probability distribution functions; mixture probability distribution functions; metaheuristic optimization methods; Mayfly algorithm; statistical error

## 1. Introduction

Renewable energy sources are increasingly being included into power systems. They can be found in a variety of sizes, either as a centralized massive power plant or as a dispersed generation close to the end-users [1]. Furthermore, a hybrid system can be used to meet a region's load requirements by combining several renewable sources [2]. When compared to a single source, combining renewable sources such as wind and solar with backup units gives a more reliable, environmentally friendly, and cost-effective load supply [3]. With over 77% of new capacity, wind and solar energy have seen the most rapid expansion in renewable energy outputs [4]. Renewable energy resources, despite being a clean and abundant source of energy, suffer from a lack of energy density and intermittency [5]. Researchers face the greatest difficulty in successfully anticipating and controlling renewable energy resources. Renewable energy generators, unlike conventional generators (such as coal or steam turbine power plants), can only generate electricity

when the renewable resources are available. As a result, the accurate forecasting, control, and representation of renewable energy systems are critical for ensuring a steady and uninterrupted energy supply [6]. Historical records of the wind speed or solar radiation data can be analyzed to precisely assess the renewable energy potentials of any place for the production of electrical energy. This is the first stage in determining where the hybrid energy system will be installed. In a wind system, the wind speed variable is the most essential parameter in wind power modeling [7,8]. Wind speed is a random variable that varies over time and is influenced by geographical and climatic factors in the area [9,10]. The solar irradiance variable is the most essential parameter in Photovoltaic (PV) generation [11].

Forecasting wind speed and solar radiation is essential for determining wind energy and photovoltaic power generation. Many PDFs are utilized to simulate wind speed and solar irradiances such as the Weibull distribution, Inverse-Gaussian distribution, Gamma distribution, Lognormal distribution, and others. Various PDFs have been proposed in the literature to simulate wind speed and solar irradiance. Many researchers have proposed the Weibull distribution in wind speed modeling [12,13]. Auwera [14] applied the three-parameters of the Weibull distribution and he claims that it can more accurately capture wind speed data than the usual two-parameter Weibull function. JA Carta [15] utilized mixed distributions to create a bi-modal framework for wind frequency distribution histograms. Celik [16] used Weibull-representative wind data instead of actual time-series data and discovered that the calculated wind energy was extremely accurate. Khamees [17] applied a mixture of distribution functions to simulate wind speed frequency distribution for a site located in the USA. Brano [18] used seven probability distributions and compared them to determine which was the greatest fit for the urban area. Labeeuw [19] used the Weibull, Gamma, Lognormal, and Rayleigh distributions to estimate the electricity usage of various residential homes incorporating PV generation. Salameh [20] provided a probabilistic generation model for solar irradiance uncertainty modeling based on the Beta distribution. Guangyuan [21] used several probabilistic models to estimate sun irradiation and evaluate their efficacy using real-world data. Arevalo [22] utilized Lognormal distribution for the solar irradiance probability distribution function to get the uncertainty cost functions for electrical systems.

The original distributions suffer from a notable error between the real readings and the estimated distribution curve; therefore, this work presents an improvement in the distribution curves by introducing the two and three-component mixture distributions between the four original distributions. The Weibull, Lognormal, Gamma, and Inverse-Gaussian probability distribution functions are presented in this paper to simulate wind speed and solar irradiance frequency distributions. The two-component mixtures—the Weibull–Gamma distribution, the Weibull–Lognormal distribution, the mixture of the Weibull–Inverse-Gaussian distribution, the mixture of the Lognormal distribution, the mixture of the Weibull distribution, the mixture of the Gamma distribution, and the mixture of the Inverse-Gaussian distribution—are all proposed in this work to improve the fitting of wind speed and solar irradiance. Moreover, to get more improvement in the simulation of wind speed and solar irradiance, the three-component mixtures of the Weibull distributions—the Weibull–Weibull–Gamma and the Weibull–Gamma–Gamma—are presented. Three statistical errors are used to test and compare all distributions. The correlation coefficient ($R^2$), chi-square ($X^2$), and root mean square error (RMSE) were employed in this study as statistical errors. The results show that the two-component mixture distributions improve data fitting more than the four original distributions, and the three-component mixture distributions improve data fitting more than the two-component mixture distributions and provide the best fit for the wind speed and solar irradiance being studied. This work proposed a novel metaheuristic method called the Mayfly algorithm [23] to optimize the parameters of the original and the mixture of PDFs. This method's goal is to reduce the RMSE between the real data and the simulated data from the distribution curve. The performance of the MA method is better than the numerical methods in terms of finding

the parameters of the PDFs [8]. This paper utilizes wind speed and solar irradiations' data collected from locations in the USA for 5 years [24,25].

The remainder of this paper is arranged as follows: Section 2 presents the four original distribution functions utilized to simulate wind speed and solar irradiance, the two and three-component mixtures of PDFs, the statistical errors utilized in this study, and the Mayfly algorithms. The results and discussion of the wind speed simulation are described in Section 3. Section 4 presents the results and discussion of the solar irradiance simulation. Finally, the conclusion is presented in Section 5.

## 2. Mathematical Modeling

### *2.1. The Original Probability Distribution Functions*

There are a variety of statistical distribution functions that can be used to model the behavior of wind speed and solar irradiance data over time; this paper will cover the most well-known four distributions.

#### 2.1.1. The Weibull Distribution

The Weibull PDF was developed by W. Weibull [26]. The probability density function $f_w(x, k, c)$ is given by:

$$f_w(x, k, c) = \frac{k}{c^k} x^{x-1} \exp\left(-\left(\frac{x}{c}\right)^k\right)$$
(1)

where the shape and scale parameters are $k$ and $c$, respectively.

#### 2.1.2. The Lognormal Distribution

The Lognormal distribution is a type of normal distribution that includes a random variable, where the logarithm has a normal distribution [27].

The Lognormal distribution $f_l(x, \alpha, \beta)$ is given by:

$$f_l(x, \alpha, \beta) = \frac{1}{x\beta\sqrt{2\pi}} \exp\left(-\frac{1}{2}\left(\frac{\ln x - \alpha}{\beta}\right)^2\right)$$
(2)

where $\alpha$ and $\beta$ are the mean and standard deviation, respectively.

#### 2.1.3. The Gamma Distribution

The Gamma distribution models the sums of exponentially distributed random variables and is a generalization of the chi-square and exponential distributions. [28].

The Gamma distribution $f_G(x, a, b)$ is given by:

$$f_G(x, a, b) = \frac{1}{b^a \, \Gamma(a)} \, x^{a-1} \, e^{-\frac{x}{b}}$$
(3)

where the shape and scale parameters are $a$ and $b$, respectively.

#### 2.1.4. The Inverse-Gaussian Distribution

To describe nonnegative positively skewed data, the Inverse-Gaussian Distribution is utilized. In many ways, it mimics standard Gaussian (normal) distributions, making it helpful in inferential statistics [29].

The Inverse-Gaussian distribution $f_{IG}(x, \lambda, \mu)$ is given by:

$$f_{IG}(x, \lambda, \mu) = \sqrt{\frac{\lambda}{2\pi x^3}} \exp\left(-\frac{\lambda \, (x - \mu)^2}{2\mu^2 x}\right)$$
(4)

where the shape and scale parameters are $\lambda$ and $\mu$, respectively.

### 2.2. The Two-Component Mixture of Probability Distribution Functions

The two-component mixture of probability distribution functions is an integration of two PDFs from the same type or from different types for which each PDF is given a specific weight ($w$). This work presents seven two-component mixtures of distribution functions generated from various integrations of the Weibull, Gamma, Lognormal, and Inverse-Gaussian distributions.

#### 2.2.1. The Two-Component Mixture of the Weibull Distribution

The mixture of the Weibull distribution $f_{2w}(x, k, c)$ is made up of two Weibull distributions, each with different scale and shape parameters and a different weight applied to each distribution function:

$$f_{2w}(x, k, c) = w * f_w(x, k1, c1) + (1 - w) * f_w(x, k2, c2) \tag{5}$$

where $0 \leq w \leq 1$.

#### 2.2.2. The Two-Component Mixture of the Gamma Distribution

The mixture of the Gamma distribution $f_{2G}(v, a, b)$ is made up of two Gamma distributions, each with different parameters and a different weight applied to each distribution function:

$$f_{2G}(x, a, b) = w * f_G(x, a1, b1) + (1 - w) * f_G(x, a2, b2) \tag{6}$$

#### 2.2.3. The Two-Component Mixture of the Lognormal Distribution

The mixture of the Lognormal distribution $f_{2L}(v, \alpha, \beta)$ is made up of two Lognormal distributions, each with different parameters and a different weight applied to each distribution function:

$$f_{2L}(x, \alpha, \beta) = w * f_{L2}(x, \alpha1, \beta1) + (1 - w) * f_{L2}(x, \alpha2, \beta2) \tag{7}$$

#### 2.2.4. The Two-Component Mixture of the Inverse-Gaussian Distribution

The mixture of the Inverse-Gaussian distribution $f_{2IG}(v, \lambda, \mu)$ is made up of two Inverse-Gaussian distributions, each with different parameters and a different weight applied to each distribution function:

$$f_{2IG}(x, \lambda, \mu) = w * f_{IG}(x, \lambda1, \mu1) + (1 - w) * f_{IG}(x, \lambda2, \mu2) \tag{8}$$

#### 2.2.5. The Two-Component Mixture of the Weibull–Gamma Distribution

The PDF of the mixture of the Weibull–Gamma distribution $f_{w-G}(x, k, c, a, b)$ can be written as:

$$f_{w-G}(x, k, c, a, b) = w * f_w(x, k, c) + (1 - w) * f_G(x, a, b) \tag{9}$$

#### 2.2.6. The Two-Component Mixture of the Weibull–Lognormal Distribution

The PDF of the mixture of the Weibull–Lognormal distribution $f_{w-L}(x, k, c, \alpha, \beta)$ can be written as:

$$f_{w-G}(x, k, c, \alpha, \beta) = w * f_w(x, k, c) + (1 - w) * f_l(x, \alpha, \beta) \tag{10}$$

#### 2.2.7. The Two-Component Mixture of the Weibull–Inverse-Gaussian Distribution

The PDF of the mixture of the Weibull–Inverse-Gaussian distribution $f_{w-IG}(x, k, c, \lambda, \mu)$ can be written as:

$$f_{w-IG}(x, k, c, \lambda, \mu) = w * f_w(x, k, c) + (1 - w) * f_{IG}(x, \lambda, \mu) \tag{11}$$

### 2.3. The Three-Component Mixture of Probability Distribution Functions

The three-component mixture of probability distribution functions is an integration of three PDFs from the same type or from different types for which each PDF is given a specific weight $(w)$. This study presents three three-component mixtures of distribution functions generated from the integration of the Weibull and Gamma distributions.

#### 2.3.1. The Three-Component Mixture of the Weibull Distribution

The mixture of the Weibull distribution $f_{3w}(x, k, c)$ is made up of three families of Weibull distributions, each with a different scale and shape parameter and a different weight applied to each distribution function:

$$f_{3w} = w1 * f_w(x, k1, c1) + w2 * f_w(x, k2, c2) + w3 * f_w(x, k3, c3) \tag{12}$$

where $w1 + w2 + w3 = 1$.

#### 2.3.2. The Three-Component Mixture of the Weibull–Weibull–Gamma Distribution

The PDF of the mixture of the Weibull–Weibull–Gamma distribution $f_{W-W-G}$ can be written as:

$$f_{W-W-G} = w1 * f_w(x, k1, c1) + w2 * f_w(x, k2, c2) + w3 * f_G(x, a, b) \tag{13}$$

#### 2.3.3. The Three-Component Mixture of the Weibull–Gamma–Gamma Distribution

The PDF of the mixture of the Weibull–Gamma–Gamma distribution $f_{W-G-G}$ can be written as:

$$f_{W-G-G} = w1 * f_w(x, k, c) + w2 * f_G(x, a1, b1) + w3 * f_G(x, a2, b2) \tag{14}$$

### 2.4. Statistical Error Anaylsis

Statistical error techniques are used to examine and compare the performance and accuracy of the distribution functions provided in this work. The statistical error procedures that were employed are the root mean square error (RMSE), chi-square error ($X^2$), and correlation coefficient ($R^2$). The following is a summary of these [30]:

$$RMSE = \sqrt{\frac{1}{n} \sum_{i=1}^{n} (y_i - x_i)^2} \tag{15}$$

$$R^2 = \frac{\sum_{i=1}^{n} (y_i - m)^2 - \sum_{i=1}^{n} (x_i - m)^2}{\sum_{i=1}^{n} (y_i - m)^2} \tag{16}$$

$$X^2 = \frac{1}{N - n} \sum_{i=1}^{n} (y_i - x_i)^2 \tag{17}$$

where $n$ is the number of frequency distribution classes, $N$ is the number of observations, $y_i$ is the real data from the site, $x_i$ is the estimated data from the distribution function, and m is the average of the collected data.

### 2.5. The Mayfly Algorithm

One method to get the best solutions to problems such as those outlined in this paper is to use artificial intelligence optimization algorithms [31]. The majority of these algorithms are based on natural events, with an objective function serving as a distance requirement for the best solutions. The selection of the goal function has the most impact on the results. The Mayfly algorithm [23] is one of the artificial intelligence optimization algorithms.

The mayfly is a type of insect that belongs to the Balaenoptera family. Mayflies emerge as aquatic nymphs from their eggs, then ascend to the surface when fully grown, where they live for only a few days before reproducing and dying. To mate with a Female Mayfly (FM),

a Male Mayfly (MM) performs a nuptial dance movement around a water body; FMs mate with MMs in the air and eventually drop offspring/eggs, and the life cycle continues. In this study, the populations in the search space are representing the distribution parameters and the weight of each distribution in the mixture of components.

The Mayfly algorithm can be divided into four categories as follows:

### 2.5.1. The Males' Movement

Swarms of MMs assemble around a body of water. This means that their position and movement speed change in reaction to the other mayflies in the swarm. The following is how MMs can be described:

$$x_i^{t+1} = x_i^t + v_i^{t+1} \tag{18}$$

where $x$ and $v$ are the position and velocity of the Male Mayflies, respectively.

The velocity of MMs can be calculated as:

$$v_i^{t+1} = v_i^t + ae^{-\beta r_p^2}\left(pbest_i - x_i^t\right) + be^{-\beta r_g^2}\left(gbest - x_i^t\right) \tag{19}$$

where $a$ and $b$ are constants; *pbest* and *gbest* are the local and global best positions, respectively; the distance between the current position and best position is $r_p$; the distance between the current position and the global best position is $r_g$.

In this study, $xi$ represents the distribution parameters and the weight of each distribution in the mixture of components, and pbest and gbest are representing the local and global values of RMSE, respectively.

### 2.5.2. The Females' Movement

Females do not cluster in swarms, they move to the male's position to reproduce. To calculate the change in this position, use the equations below:

$$y_i^{t+1} = y_i^t + w_i^{t+1} \tag{20}$$

where $y$ and $w$ are the position and velocity of the Female Mayflies, respectively.

In this study, $yi$ represents the distribution parameters and the weight of each distribution in the mixture components.

The velocity of FM can be calculated as:

$$w_i^{t+1} = w_i^t + be^{-\beta r_m^2}\left(x_i^t - y_i^t\right) \tag{21}$$

where $r_m$ is the distance between the Males and Females.

### 2.5.3. Mating

The offspring are chosen in the same way that the Females choose their breeding Males. The best Male couples with the best Female to develop and produce progeny. The rankings of all the Males and Females are the same. The MA crossover is calculated using the following equations:

$$offspring1 = L \times x_i + (1 - L) \times y_i \tag{22}$$

$$offspring2 = L \times y_i + (1 - L) \times x_i \tag{23}$$

where $L$ is a random value $0 \leq L \leq 1$.

### 2.5.4. Mutation

The offspring are altered to prevent the process from becoming stuck on a local minimum, which may be expressed as the equation:

$$offspring_n' = offspring_n + N_{(0,1)} \tag{24}$$

where $N_{(0,1)}$ is the Normal PDF with $\mu = 0$ and $\sigma = 1$.
The flowchart of the MA is shown in Figure 1.

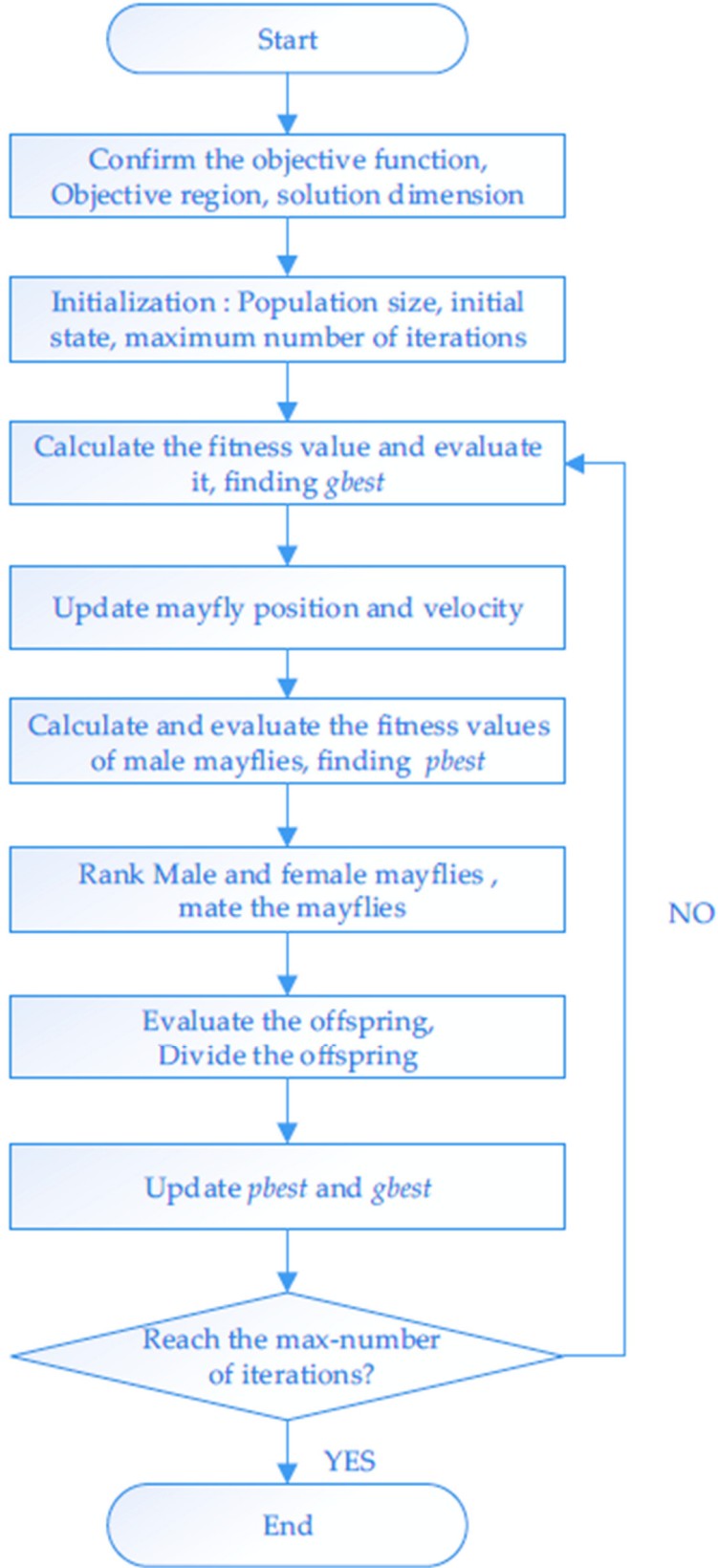

**Figure 1.** Flowchart of the Mayfly algorithm.

## 3. Wind Speed Modelling

Hourly mean wind speed (m/s) data obtained from a wind farm in the USA for 5 years [24] was used for the modelling. This study assesses the different original and mixed probability distribution functions to test their usefulness to simulate wind speed. The original PDFs used here are the Weibull, Lognormal, Gamma, and Inverse-Gaussian distribution functions. Combinations of PDFs from various types or the same types constitute the mixture distributions. This study compares the performance between the original, the two-component mixture, and the three-component mixture probability distribution functions using three statistical errors. The MA technique is used to estimate the PDFs' parameters, and the goal of the MA is to minimize the RMSE.

### 3.1. The Original Probability Distribution Functions

Four original PDFs are presented in this case. Table 1 shows the statistical errors and the distribution parameters for the four PDFs by using the MA approach. The Gamma distribution produces the best results because it has the lowest RMSE and $X^2$ and the highest $R^2$. Figure 2 shows the fitting of the four original PDFs with wind speed frequency distribution where it is evident that the Gamma distribution has the best fitting. The Gamma distribution decreases the RMSE by 38%, 74%, and 45% compared to the Weibull, Lognormal, and Inverse-Gaussian distributions, respectively.

**Table 1.** The probability distribution parameters for original PDFs in wind speed modeling.

| Mixture Distribution Function | Parameters | RMSE | $R^2$ | $X^2$ |
|---|---|---|---|---|
| Weibull | $k = 1.7253$ <br> $c = 3.3935$ | 0.00624 | 0.99289 | 0.00021 |
| Lognormal | $\alpha = 1.1559$ <br> $\beta = 1$ | 0.02495 | 0.80767 | 0.00336 |
| Gamma | $a = 2.6061$ <br> $b = 1.2364$ | 0.00384 | 0.99728 | 0.00007 |
| Inverse-Gaussian | $\mu = 3.6769$ <br> $\lambda = 6.8869$ | 0.00707 | 0.99027 | 0.00027 |

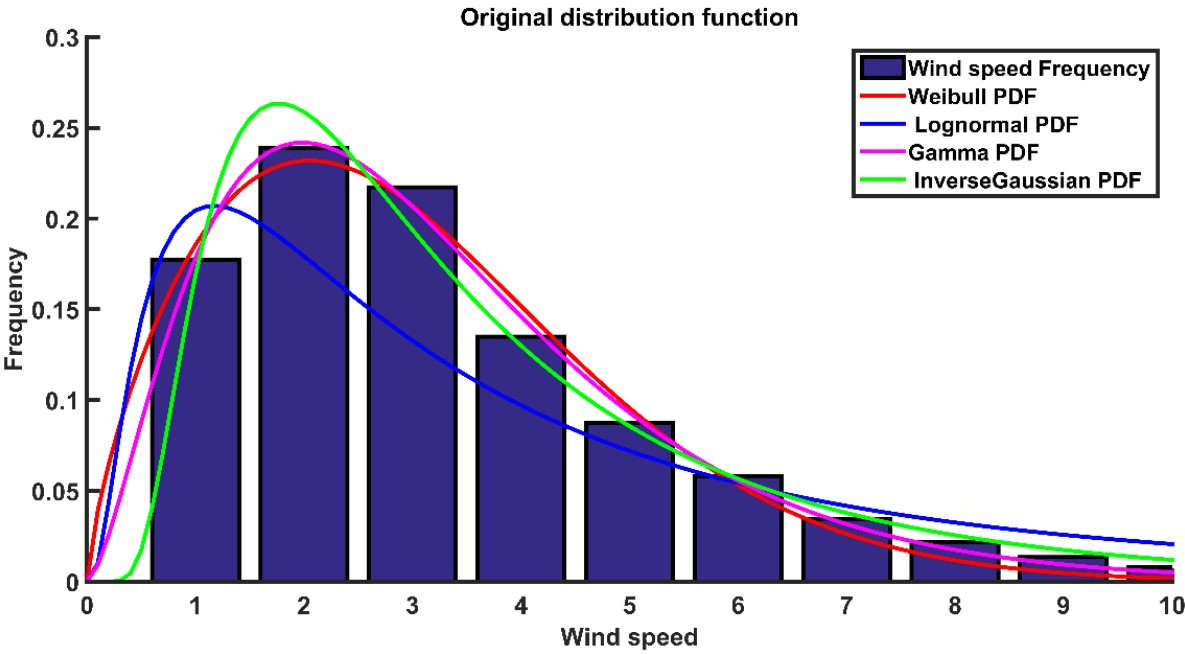

**Figure 2.** Original distribution functions in wind speed modeling.

### 3.2. The Two-Component Mixtures of Probability Distribution Functions

In this case, seven two-component mixture distribution functions were studied to indicate the best curve that fits the wind speed frequency distribution. Table 2 shows the seven mixtures of PDFs being studied combined with their optimal parameters and statistical error calculations. The two-component mixtures, the Weibull and the mixture of the Weibull–Gamma PDFs, produce the best results because they have the lowest RMSE and $X^2$ and the highest $R^2$. In addition, the results show that the statistical error calculations of the two-component mixture of PDFs are better than the four original distribution functions. Figure 3 shows the fitting of the seven two-component mixtures of PDFs to the wind speed frequency distribution. It is notable that the two-component mixtures of the Weibull and the Weibull–Gamma decrease the RMSE by 32%, and 30% respectively, when compared to the Gamma distribution.

**Table 2.** Probability distribution parameters for the two-component mixture PDFs in the wind speed modeling.

| Mixture Distribution Function | Parameters | RMSE | $R^2$ | $X^2$ |
|---|---|---|---|---|
| Two-component Mixture of Weibull | $C1 = 5.934$<br>$K1 = 2.553$<br>$C2 = 2.800$<br>$K2 = 1.970$<br>$W = 0.265$ | 0.002599 | 0.99873 | 0.00003 |
| Two-component Mixture of Gamma | $a1 = 2.9609\ b1 = 1$<br>$a2 = 2.083$<br>$b2 = 1.982$<br>$W = 0.689$ | 0.0033555 | 0.99788 | 0.00006 |
| Two-component Mixture of Lognormal | $\alpha1 = 1.155$<br>$\beta1 = 1$<br>$\alpha2 = 1.155\ \beta2 = 1$<br>$W = 0.203$ | 0.02495 | 0.80767 | 0.0033685 |
| Two-component Mixture of Inverse-Gaussian | $\mu1 = 1$<br>$\lambda1 = 6.128$<br>$\mu2 = 3.704$<br>$\lambda2 = 10$<br>$W = 0.1$ | 0.0029649 | 0.99835 | 0.0000475 |
| Weibull–Gamma Mixture | $k1 = 1.9781$<br>$C1 = 2.5913$<br>$a2 = 5.0456$<br>$b2 = 1$<br>$W = 0.31$ | 0.0026499 | 0.99867 | 0.000037 |
| Weibull–Lognormal Mixture | $k = 1.7371$<br>$c = 3.3862$<br>$\alpha = 4.5158$<br>$\beta = 3.5407$<br>$W = 0.9927$ | 0.0062237 | 0.9929 | 0.0002096 |
| Weibull–Inverse-Gaussian Mixture | $K = 1.7826$<br>$C = 3.2469$<br>$\mu = 4.0216$<br>$\lambda = 6.9001$<br>$W = 0.5695$ | 0.0039661 | 0.99702 | 0.00008 |

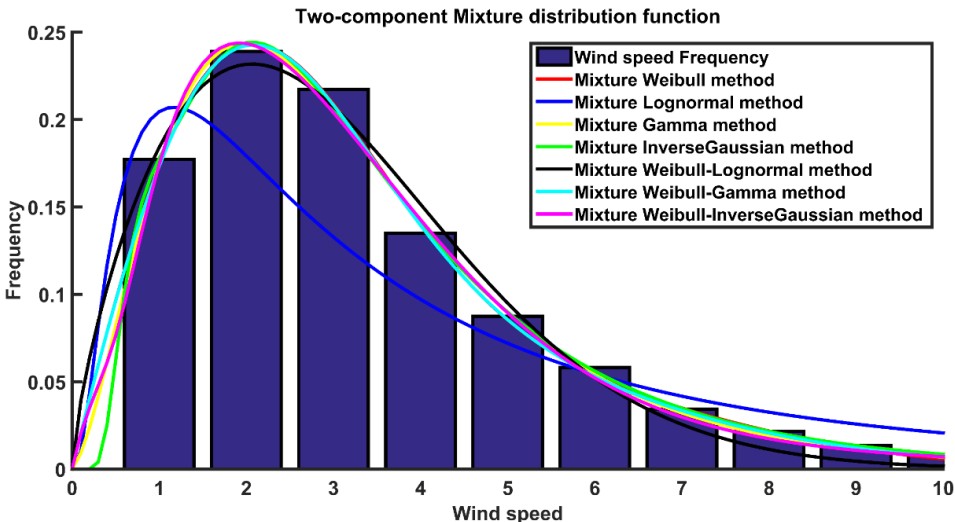

**Figure 3.** The two-component mixture distribution functions in wind speed modeling.

### 3.3. The Three-Component Mixtures of Probability Distribution Functions

To get more improvement in the calculations of the statistical errors, the three-component mixtures of PDFs were used to simulate wind speed frequency distribution. As indicated in the previous section, the two-component mixtures of the Weibull and Weibull–Gamma PDFs have the best fitting criteria for wind data among the original and two-component mixture distributions. Therefore, in this case, we study the three-component mixtures of the Weibull, Weibull–Weibull–Gamma, and Weibull–Gamma–Gamma. Table 3 shows the three-component mixtures of the PDFs being studied combined with their optimal parameters and statistical error calculations. Figure 4 shows the fitting of the three-component mixtures of PDFs to the wind speed frequency distribution. The three-component mixtures of the Weibull–Gamma–Gamma PDFs produces the best results because it has the lowest RMSE and $X^2$ and the highest $R^2$. In addition, the results show that the statistical error calculations of the three-component mixture PDFs are better than the four original distributions and the seven two-component mixtures of the distribution functions being studied. From the results, it is apparent that the three-component mixture of the Weibull decreases the RMSE by 71%, 57%, and 58% when compared to the Gamma, the two-component Weibull and the *Weibull–Gamma mixture* distributions, respectively.

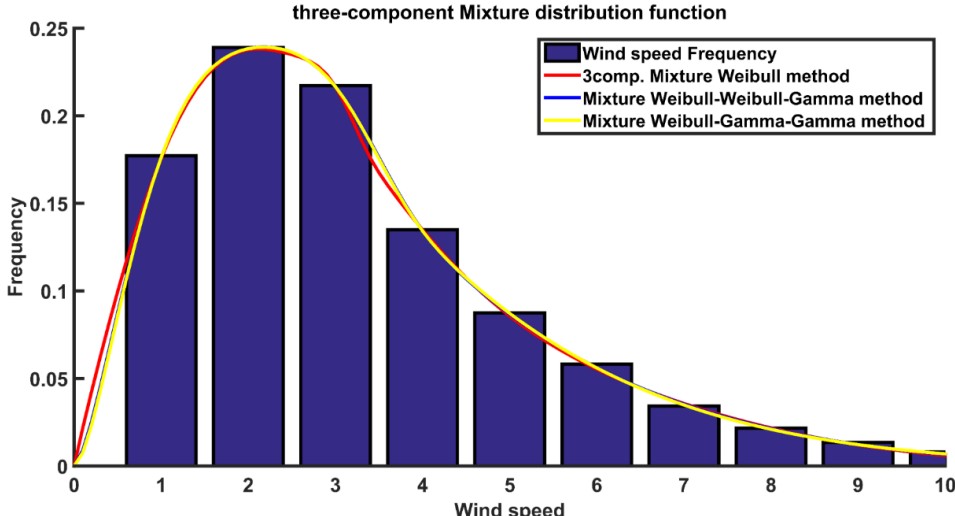

**Figure 4.** The three-component mixture distribution functions in wind speed modeling.

**Table 3.** Probability distribution parameters for the three-component mixtures of PDFs in wind speed modeling.

| Mixture Distribution Function | Parameters | RMSE | $R^2$ | $X^2$ |
|---|---|---|---|---|
| Three-component Mixture of Weibull | $C1 = 5.2195$<br>$K1 = 2.1407$<br>$C2 = 2.6216$<br>$K2 = 1.9431$<br>$C3 = 2.9419$<br>$K3 = 10$<br>$W1 = 0.39092$<br>$W2 = 0.59452$ | 0.0010965 | 0.99977 | 0.0000065 |
| Three-component Mixture of Weibull–Weibull–Gamma | $C1 = 5.336$<br>$K1 = 2.0162$<br>$C2 = 3.0395$<br>$K2 = 5.2403$<br>$a = 2.7292$<br>$b = 1.0015$<br>$W1 = 0.27047$<br>$W2 = 0.05$ | 0.00073657 | 0.9999 | 0.0000029 |
| Three-component Mixture of Weibull–Gamma–Gamma | $C1 = 3.0208$<br>$K1 = 5.2757$<br>$a1 = 5.6276$<br>$b1 = 1$<br>$a2 = 2.7443$<br>$b2 = 1$<br>$W1 = 0.05$<br>$W2 = 0.1904$ | 0.00068585 | 0.99991 | 0.0000025 |

## 4. Solar Irradiation Modelling

Using daily solar irradiance $(\text{kwh/m}^2)$ obtained from a solar farm in the USA for 5 years [25], this study examines the suitability of several original and mixed probability distribution functions for simulating solar irradiation. The Weibull, Lognormal, Gamma, and Inverse-Gaussian distribution functions are examined as original PDFs. Combinations of the PDFs from the various types or the same types constitute the mixture distributions. Using three statistical errors, this study examines the performance of the original, the two-component mixtures, and three-component mixtures of probability distribution functions. The MA technique is used to estimate the PDFs' parameters, and the goal of the MA is to minimize the RMSE.

### 4.1. Original Probability Distribution Functions

In this case, four original PDFs are shown. Table 4 shows the statistical errors and the PDFs' parameters for the four PDFs using the MA approach. The Weibull distribution gives the best outcomes because it has the lowest RMSE and $X^2$ and the highest $R^2$. The fitting of the four original PDFs with the solar irradiance frequency distribution is shown in Figure 5, with the Weibull distribution having the best fit. It is evident that the Weibull distribution decreases the RMSE by 7%, 10%, and 11% compared with the Gamma, Lognormal, and Inverse-Gaussian distributions, respectively.

### 4.2. The Two-Component Mixtures of Probability Distribution Functions

Seven two-component mixtures of distribution functions are investigated in this scenario to determine which curve best fits the solar irradiance frequency distribution. Table 5 shows the seven mixtures of the PDFs being studied combined with their optimal parameters and statistical error calculations. The two-component mixture of the Weibull and the Weibull–Gamma mixture of PDFs produces the best results because it has the lowest RMSE and $X^2$ and the highest $R^2$. Furthermore, the results demonstrate that the statistical

error estimations of the two-component mixtures of PDFs are superior to the four original distribution functions. Figure 6 shows the fitting of the seven two-component mixtures of PDFs to the solar irradiance frequency distribution. It is evident that the two-component mixture of the Weibull and the *Weibull–Gamma mixture* decreases the RMSE by 67%, and 70%, respectively, when compared to the Weibull distribution.

**Table 4.** Probability distribution parameters for the original PDFs in the solar irradiance modeling.

| Mixture Distribution Function | Parameters | RMSE | $R^2$ | $X^2$ |
|---|---|---|---|---|
| Weibull | $k = 4.0832$ <br> $c = 10.7769$ | 0.046732 | 0.29587 | 0.48938 |
| Lognormal | $\alpha = 2.319$ <br> $\beta = 0.31785$ | 0.052494 | 0.16719 | 0.61751 |
| Gamma | $a = 10.968$ <br> $b = 0.9481$ | 0.050583 | 0.20968 | 0.57335 |
| Inverse-Gaussian | $\mu = 10.6831$ <br> $\lambda = 101.55$ | 0.052646 | 0.16171 | 0.62109 |

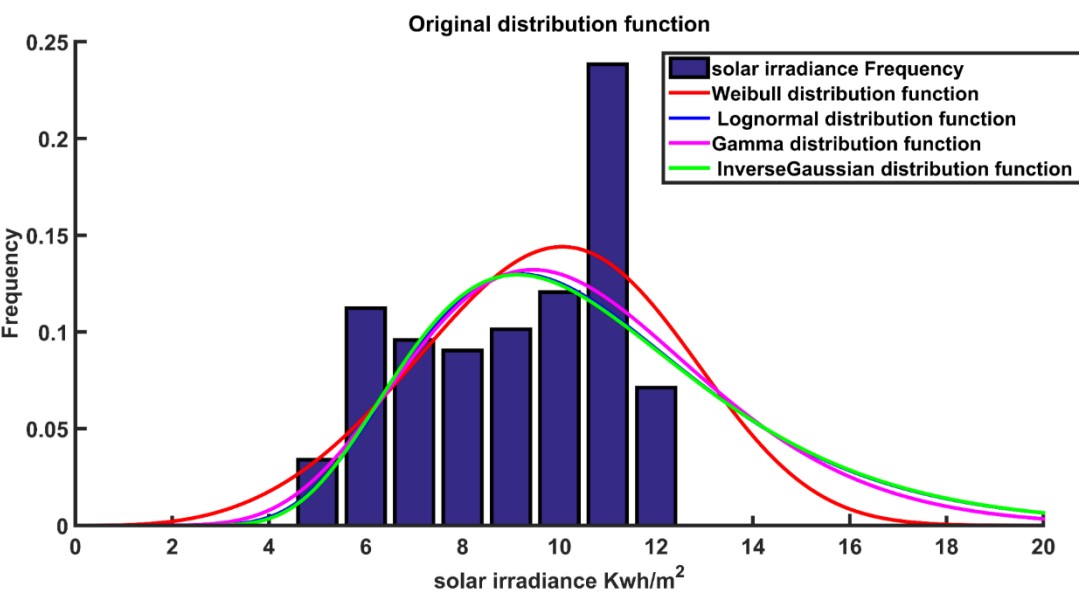

**Figure 5.** The original distribution function in the solar irradiance modeling.

*4.3. The Three-Component Mixtures of Probability Distribution Functions*

The three-component mixtures of PDFs are utilized to model solar irradiance frequency distribution in order to enhance the statistical error calculations. Among the original and two-component mixture distributions, the two-component mixture of the Weibull and Weibull–Gamma PDFs has the best fitting criteria for solar irradiance, as shown in the previous section. Therefore, in this case, we study the three-component mixtures of the Weibull, Weibull–Weibull–Gamma, and Weibull–Gamma–Gamma. Table 6 shows the three-component mixture of the PDFs being studied combined with their optimal parameters and statistical error calculations. Figure 7 shows the fitting of the three-component mixtures of PDFs to the wind speed frequency distribution. The three-component mixture of the Weibull–Gamma–Gamma PDF gives the best outcomes because it has the lowest RMSE and $X^2$ and the highest $R^2$. The results also reveal that the statistical error calculations of the three-component mixtures of PDFs are better than the four original distributions and the seven two-component mixtures of distribution functions. From the results, it is evident that the three-component mixture of Weibull–Gamma–Gamma decreases the RMSE by 73%, 17%,

and 9%, when compared to the distributions of the Weibull, the two-component Weibull, and the Weibull–Gamma mixture, respectively.

**Table 5.** Probability distribution parameters for the two-component mixtures of PDFs in the solar irradiance modeling.

| Mixture Distribution Function | Parameters | RMSE | $R^2$ | $X^2$ |
|---|---|---|---|---|
| Two-component Mixture of Weibull | $C1 = 11.259$<br>$K1 = 21.903$<br>$C2 = 8.8788$<br>$K2 = 3.2454$<br>$W = 0.2777$ | 0.015156 | 0.92509 | 0.051472 |
| Two-component Mixture of Gamma | $a1 = 25$<br>$b1 = 0.2559$<br>$a2 = 25$<br>$b2 = 0.4382$<br>$W = 0.2464$ | 0.041374 | 0.44749 | 0.38359 |
| Two-component Mixture of Lognormal | $\alpha1 = 1.8872$<br>$\beta1 = 0.1588$<br>$\alpha2 = 2.383$<br>$\beta2 = 0.1416$<br>$W = 0.3$ | 0.034767 | 0.60672 | 0.27087 |
| Two-component Mixture of Inverse-Gaussian | $\mu1 = 10.948$<br>$\lambda1 = 120$<br>$\mu2 = 10$<br>$\lambda2 = 64.655$<br>$W = 0.6760$ | 0.052049 | 0.18665 | 0.60707 |
| Weibull–Gamma Mixture | $k1 = 21.457$<br>$C1 = 11.224$<br>$a2 = 8.676$<br>$b2 = 0.993$<br>$W = 0.2747$ | 0.013752 | 0.93832 | 0.042379 |
| Weibull–Lognormal Mixture | $k = 3.7122$<br>$c = 9.8743$<br>$\alpha = 2.3805$<br>$\beta = 0.01052$<br>$W = 0.9650$ | 0.042295 | 0.41596 | 0.40086 |
| Weibull–Inverse-Gaussian Mixture | $K = 13.3938$<br>$C = 11.324$<br>$\mu = 11.489$<br>$\lambda = 31.870$<br>$W = 0.4073$ | 0.042353 | 0.45024 | 0.40197 |

**Table 6.** Probability distribution parameters for the three-component mixtures of PDFs in the solar irradiance modeling.

| Mixture Distribution Function | Parameters | RMSE | $R^2$ | $X^2$ |
|---|---|---|---|---|
| Three-component Mixture of Weibull | $C1 = 6.9165$<br>$K1 = 5.1352$<br>$C2 = 10.7532$<br>$K2 = 4.1362$<br>$C3 = 11.1145$<br>$K3 = 20$<br>$W1 = 0.24226$<br>$W2 = 0.5$ | 0.013409 | 0.94138 | 0.04029 |

**Table 6.** *Cont.*

| Mixture Distribution Function | Parameters | RMSE | $R^2$ | $X^2$ |
|---|---|---|---|---|
| Three-component Mixture of Weibull–Weibull–Gamma | $C1 = 11.1816$<br>$K1 = 17.0393$<br>$C2 = 7.8099$<br>$K2 = 4.33$<br>$a = 15.4322$<br>$b = 17.262$<br>$W1 = 0.41269$<br>$W2 = 0.5$ | 0.014118 | 0.93492 | 0.044663 |
| Three-component Mixture of Weibull–Gamma–Gamma | $C = 11.1678$<br>$K = 17.0756$<br>$a1 = 15.6347$<br>$b1 = 0.48253$<br>$a2 = 15.4617$<br>$b2 = 8.3484$<br>$W1 = 0.39194$<br>$W2 = 0.49882$ | 0.012515 | 0.94886 | 0.035096 |

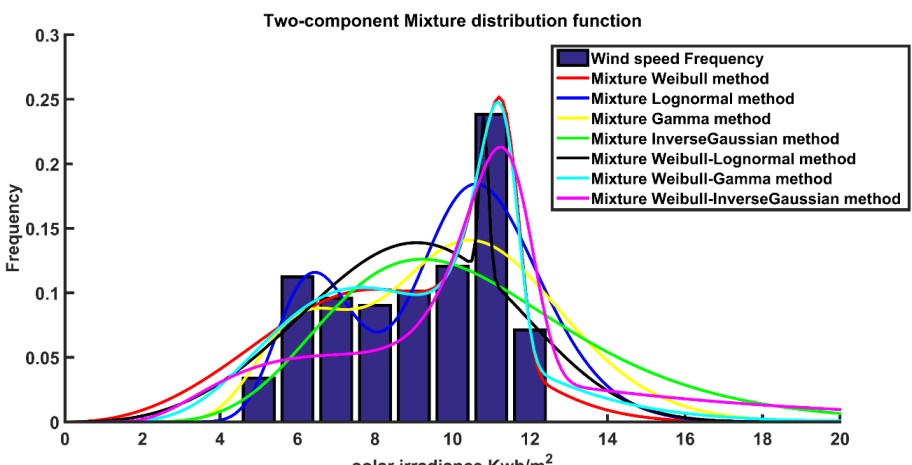

**Figure 6.** The two-component mixture distribution functions in the solar irradiance modeling.

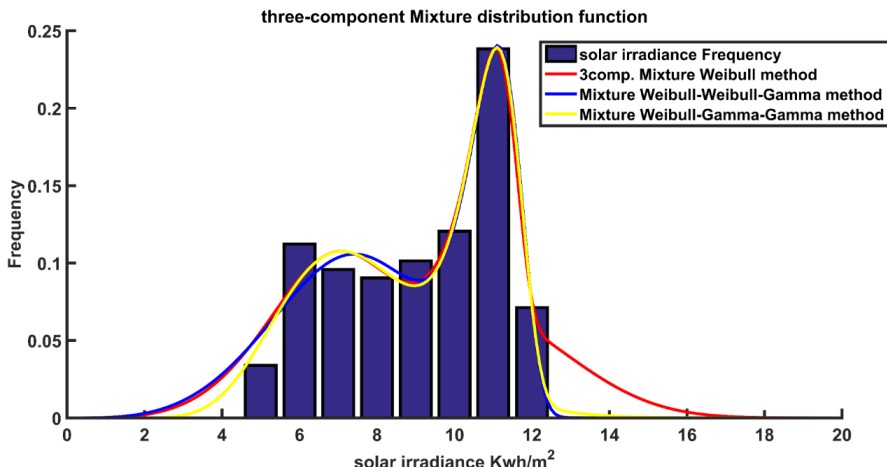

**Figure 7.** The three-component mixture distribution functions in the solar irradiance modeling.

## 5. Conclusions

This study presents an enhancement of the most famous four original PDFs, which are used to simulate wind speed and solar irradiance frequency distribution by using two

and three-component mixtures of distribution functions. The RMSE is used as an objective function that must be minimized, and the novel metaheuristic Mayfly algorithm is used to estimate the parameters of the distribution functions. The weights of mixture distribution and the parameters of the distribution functions are the optimization algorithm's control variables. This study analyses 17 distinct distribution functions for simulating wind speed and solar irradiance. Original and mixtures of PDFs are used in the wind speed and solar irradiance modeling. Original PDFs such as the Weibull, Lognormal, Inverse-Gaussian, and Gamma are proven to be insufficient; therefore, the mixtures of distribution functions are utilized to better simulate the observed wind speed and solar irradiance data. The results show that the two-component mixtures of distributions fit the wind and solar irradiance data better and have lower statistical errors than the original distribution. The three-component mixtures of distributions have the best statistical error calculations when compared to the original and the two-component distributions. In simulating wind speed and solar irradiance frequency distributions, the results show that employing mixtures of distributions is better than using the original distribution. A mixture of PDFs, particularly the three-component mixture of the Weibull–Gamma–Gamma distribution, can give a tight fit to wind speed and solar irradiance frequency distributions, as it reduces the RMSE by 82% for wind speed modeling and 75% for solar irradiance compared to the original Gamma distribution function.

**Author Contributions:** A.K.K. and M.A.A. designed the problem under study, performed the simulations and obtained the results; M.R.E. and A.O.B. analyzed the obtained results. A.K.K. and M.A.A. wrote the paper, which was further reviewed by A.Y.A. All authors have read and agreed to the published version of the manuscript.

**Funding:** This research received no external funding.

**Data Availability Statement:** Not applicable.

**Conflicts of Interest:** The authors declare no conflict of interest.

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
