# Peer review of "The Mixture of Probability Distribution Functions for Wind and Photovoltaic Power Systems Using a Metaheuristic Method"

_processes, doi:10.3390/pr10112446_

Round 1
Reviewer 1 Report
1. The Abstract is presented very nicely. The incorporation of the quantitative results at the end would add value to it.
2. The research gap is missing from the introduction section.
3. Presenting the methodology with an illustrative flowchart or diagram is advisable.
4. Sections 2, 3, 4, 5, and 6 can be merged as a single section titled 'methodology' or 'mathematical modeling'.
5. A quantitative Improvement in wind speed and solar irradiance prediction should also be discussed along with the presently given RMSE values.
Reviewer 2 Report
Some minor typos were found. A complete revision of the English language should take care of it.
Reviewer 3 Report
Please see my comments in the pdf file.

Reviewer 4 Report
Based on four typical probability distribution functions, this paper aims to improve the sufficiency and accuracy of the modelling to predict the distribution curve. This paper combined two or three from the four original functions and then compared the modelling results. My technical concerns are:
1. The modelling results and the fitting parameters in this pater were based on the real historical data using Mayfly algorithm. However, there are no details on how the multiple parameters were calculated by Mayfly algorithm and how the real data were used.
2. In every model presented in this paper, more details should be given to explain the meaning of the multiple parameters.
3. Authors combined two or three of the four original models with a different scale and weight to each distribution functions. How the scale and weight were calculated?
4. The modelling results have shown that three-component mixture distributions have the best statistical error calculations when compared to the original and two-component distributions. However, there are four combinations from different types of the original functions, and much more from mixture of same type and different type. How was the best combination was optimized?
5. How many real data in total was collected and used in the statistic analysis in this paper?
Round 2
Reviewer 1 Report
The authors have incorporated the suggestions satisfactorily.
The manuscript is recommended for Acceptance
Reviewer 3 Report
The authors addressed most of my concerns from the first review.
